# Identifying Health Equity Factors That Influence the Public’s Perception of COVID-19 Health Information and Recommendations: A Scoping Review

**DOI:** 10.3390/ijerph191912073

**Published:** 2022-09-23

**Authors:** Shahab Sayfi, Ibrahim Alayche, Olivia Magwood, Margaret Gassanov, Ashley Motilall, Omar Dewidar, Nicole Detambel, Micayla Matthews, Rukhsana Ahmed, Holger J. Schünemann, Kevin Pottie

**Affiliations:** 1Schulich School of Medicine & Dentistry, Western University, London, ON N6A 5C1, Canada; 2Department of Biology, Faculty of Science, University of Ottawa, Ottawa, ON K1N 6N5, Canada; 3Faculty of Medicine, University of Ottawa, Ottawa, ON K1N 6N5, Canada; 4Interdisciplinary School of Health Sciences, Faculty of Health Sciences, University of Ottawa, 125 University, Ottawa, ON K1N 6N5, Canada; 5Bruyere Research Institute, University of Ottawa, 85 Primrose Avenue, Ottawa, ON K1R 6M1, Canada; 6Department of Health Research Methods, Evidence and Impact, McMaster University, 1280 Main St. W., Hamilton, ON L8S 4K1, Canada; 7Department of Communication, University at Albany—State University of New York, Albany, NY 12222, USA; 8Michael G. DeGroote Cochrane Canada and GRADE Centre, McMaster University, Hamilton, ON L8S 4K1, Canada; 9Department of Medicine, McMaster University, Hamilton, ON L8S 4K1, Canada; 10Department of Family Medicine, University of Ottawa, Ottawa, ON K1N 6N5, Canada

**Keywords:** public perception, health literacy, health equity, misinformation, COVID-19

## Abstract

The COVID-19 pandemic has impacted global public health and public trust in health recommendations. Trust in health information may waver in the context of health inequities. The objective of this scoping review is to map evidence on public perceptions of COVID-19 prevention information using the PROGRESS-Plus health equity framework. We systematically searched the MEDLINE, Cochrane Central Register of Controlled Trials, PsycInfo, and Embase databases from January 2020 to July 2021. We identified 792 citations and 31 studies published in 15 countries that met all inclusion criteria. The majority (30/31; 96.7%) of the studies used an observational design (74.2% cross-sectional, 16.1% cohort, 6.5% case study, 3.2% experimental trials). Most studies (61.3%) reported on perception, understanding, and uptake, and 35.5% reported on engagement, compliance, and adherence to COVID-19 measures. The most frequently reported sources of COVID-related information were social media, TV, news (newspapers/news websites), and government sources. We identified five important equity factors related to public trust and uptake of recommendations: education and health literacy (19 studies; 61.3%), gender (15 studies; 48.4%), age (15 studies; 48.4%), socioeconomic status (11 studies; 35.5%), and place of residence (10 studies; 32.3%). Our review suggests that equity factors play a role in public perception of COVID-19 information and recommendations. A future systematic review could be conducted to estimate the impact of equity factors on perception and behavior outcomes.

## 1. Introduction

The COVID-19 pandemic has disrupted health systems and economies worldwide [1] with morbidities and mortalities [2]. Public health measures, including lockdowns and public health recommendations, have also led to various unintended outcomes, such as isolation and mental health issues, which have magnified existing health inequities [3]. Indeed, the World Health Organization has reported that health outcomes may be influenced by social determinants of health (SDH) by 30–55% [4]. Social determinants of health (SDH) are the conditions in which we are born and live, including socioeconomic status, education, employment, race, gender, and age. The World Health Organization’s Commission on Social Determinants of Health pinpoints the critical role of health equity assessments during policy and guideline development [5,6].

Socioeconomic status and several other individual characteristics such as education, gender, and rurality are commonly associated with inadequate access and poor quality of medical care [7]. To recognize the social stratifiers that are involved in health inequities, the acronym PROGRESS-Plus (which stands for Place of residence, Race/ethnicity/culture/language, Occupation, Gender, Religion, Education/health literacy/digital health literacy, Socioeconomic status, Social capital, and Plus factors such as age and disability) can be utilized as a framework to ensure that equity components are applied in the conduct, reporting, and use of research studies [3].

Health literacy, which is described as the degree to which individuals have the capacity to obtain, process, and understand basic health information and services necessary to make appropriate health decisions, might also be of relevance to adequate healthcare access [8,9,10]. Recognizing the many social contexts in which the public may encounter health information, health literacy may improve health across the life course [11]. Together with health literacy, psychological factors such as self-determination and autonomy also influence how individuals make decisions and regulate their behaviors [12]. Individuals need to access, contextualize, and understand health information and recommendations for their wellbeing, and this happens within a society influenced by globalization, political power, and other sources of bias [13,14]. Indeed, commentaries highlight the critical importance of being conscious and aware enough to choose what to pay attention to and being able to construct meaning from experience [15].

Limited health literacy might result in the receipt of lower-quality care, making inappropriate health-related decisions, and subsequent disparities in health outcomes [16]. Margaret Whitehead [17] defined health inequity as differences in health that “are not only unnecessary and avoidable but, in addition, are considered unfair and unjust.” Health inequities result from a complex range of societal, health system, and resource limitations [18,19]. Health equity and other contextual factors may influence how recommendations are understood, perceived, acted upon, or disregarded [20]. The GRADE-Equity Working Group developed guidance on incorporating equity considerations in clinical and public health guidelines [21], health equity as an outcome [22], and devising recommendations that are tailored to specific populations [20].

Amidst the scientific development of COVID-19 tests, vaccines, and treatments, there is widespread misinformation [23], increased political polarization, and public demonstrations against restrictions. Governments have had to repeatedly impose public health restrictions within a climate of ongoing uncertainty, mistrust, and lack of adherence to public health guidance [24,25]. Several tools have been developed to support the public with understanding and decision making. Specific to COVID-19, an international project collecting and assessing global COVID-19 guidelines [26] has collated more than 5500 recommendations [27] and translated some of these into plain-language recommendation summaries [28]. Trust in the information source and comprehension of information may improve the perception and uptake of health information [29]. The objective of this scoping review was to map evidence on public perceptions of COVID-19 prevention information and recommendations using the PROGRESS-Plus health equity framework to identify knowledge gaps and future health equity research agendas.

## 2. Methods

### 2.1. Protocol

We developed a scoping review protocol using guidance from the Joanna Briggs Methods Manual for Scoping Reviewers [30] and published it on the Cochrane and Campbell Equity Methods website (https://methods.cochrane.org/equity/projects/global-mental-health, accessed on 23 July 2022). The key steps included: (1) identifying the research question; (2) identifying relevant studies; (3) study selection; (4) mapping a frequency analysis of study characteristics; and (5) collating, summarizing, and reporting key content [31]. We reported our findings according to the PRISMA Scoping Review (PRISMA-ScR) checklist [32].

### 2.2. Research Question

This scoping review was guided by the question: “What health equity factors influence the public’s perception and uptake of COVID-19 health information and recommendations?” The review aimed to identify and map characteristics associated with health inequities that influence the uptake of COVID-19 prevention information and guidelines.

### 2.3. Data Sources and Search

We searched the following four databases (through Ovid) from 1 January 2020 (the month of detection of the first confirmed case of COVID-19) to 26 July 2021, without language restrictions: MEDLINE, Cochrane Central Register of Controlled Trials, APA PsycInfo, and Embase. Our search strategies reflected the principal concepts of our research question: the public perception and uptake, and COVID-19 information and recommendations. The search strategy was peer-reviewed by a librarian (VL) from the University of Ottawa with expertise in health sciences (see Appendix A). We supplemented our search by reviewing the references of included articles.

### 2.4. Eligibility Criteria

We included studies that met the following criteria: (1) targeted the general public, including adults, youth, patients, and seniors (excluding those studies focusing on physicians, researchers, health professionals, academics, and others with specialized health knowledge); (2) described the perception, engagement, adherence, and uptake of COVID-19 health information as our outcomes of interest; and (3) reported on the COVID-19 health information and recommendations and their sources (e.g., the internet, social medial, government sources). Table 1 describes the inclusion and exclusion criteria.

### 2.5. Study Selection

Search results were imported into Covidence, an online software for systematic review management [33]. Two trained reviewers (SS, IA) screened the titles and abstracts independently, and discrepancies were resolved by a third reviewer (KP). Subsequently, two reviewers (SS, IA) independently screened full texts for eligibility, and discrepancies were resolved through discussion between reviewers.

### 2.6. Data Extraction

Two reviewers independently extracted data, and conflicts were resolved through discussion. For all included studies, data were extracted for the following variables: (1) author(s) and year of publication, (2) source country(ies), (3) study design, (4) gender(s) of participants, (5) age(s) of participants, (6) COVID-19 information content (e.g., prevention and vaccination information and/or recommendations), (7) source of health information or recommendation, (8) reported health equity factor according to the PROGRESS-Plus equity framework, (9) study objectives, (10) outcome (i.e., perception, adherence, understanding, uptake of information/misinformation, and engagement in preventive behaviors), (11) key findings, and (12) conclusions.

### 2.7. Methodological Quality Appraisal

In accordance with scoping review methodology [30,31], we did not appraise the methodological quality of the studies.

### 2.8. Data Mapping and Synthesis

We digitally produced a word cloud using WordItOut [34], an online program, to display the frequency of the terms used in the titles relevant to our objectives (e.g., terms related to equity factors, perception, uptake). The most reported terms in the titles are shown with a larger font size. We then reported the frequency of study characteristics, including source country, study design, source of information, and outcomes (See Table 2). We identified reports on health equity factors using the PROGRESS-Plus framework [7]. PROGRESS-Plus is an acronym for health equity factors, which include: Place of residence, Race/ethnicity/culture/language, Occupation, Gender/Sex, Religion, Education/health literacy/digital health literacy, Socioeconomic status, Social capital and additional context specific factors such as personal characteristics concerning discrimination (age and disability), features of relationships, and time-dependent relationships [7]. We mapped the health equity factors with the outcomes of interest, including the public’s perception and uptake of COVID-19 health information and recommendations (See Table 3).

## 3. Results

### 3.1. Literature Search

Our search identified a total of 792 unique citations; of these, 746 were excluded. Forty-six studies were assessed at full-text screening, and thirty-one met all eligibility criteria (See Figure 1). The most common reason for exclusion was irrelevant outcomes for the scope of this paper (i.e., study did not report the perception, engagement, adherence, or uptake of COVID-19 health information).

### 3.2. Frequency of Popular Terms (Word Cloud)

Figure 2 presents the most popular terms used in the titles of our included articles. This figure helps to identify what proportion of the included papers outline the equity factors in their titles. The font size of the word corresponds to its frequency of use in the included studies’ titles. The larger the terms, the higher the frequency. Health literacy was the most frequent equity term and was mentioned in 11 titles of the 31 papers (11/31, 35.5%). The next most common terms were highlighting COVID-19 information, misinformation, and information sources (8/31, 25.8%). Terms such as gender, socioeconomic status, and race and ethnicity, as equity factors, were not commonly included in titles. Further details are presented in Table 2.

### 3.3. Characteristics of Included Studies

The 31 included studies were conducted in 15 different countries. A little more than a third of the studies were based in the United States of America (12 studies; 38.7%), followed by China (4 studies; 12.9%), France (3 studies; 9.7%), Germany (3 studies; 9.7%), and the United Kingdom (3 studies; 9.7%). Most studies used a quantitative research type (29 studies; 93.5%), and others used qualitative methods solely or along with quantitative approaches (5 studies; 16.1%). The majority (30/31; 96.7%) of the studies used an observational design (74.2% cross-sectional, 16.1% cohort, 6.5% case study, 3.2% experimental trials). All studies included women and men, and several studies also reported on non-binary genders. Most studies included a broad age range of participants, and 4 studies (13%) focused only on youth or the elderly. Of 31 studies, 23 studies (74.2%) reported on the COVID-19 sources of health information. The most frequent sources of health information were social media (58% of studies), TV (48% of studies), news (newspapers/news websites) (42% of studies), government sources (26% of studies), family and friends (26% of studies), and radio (26% of studies). The studies mainly reported on COVID-19 public health prevention recommendations and related health behaviors (21 studies; 67.7%) and COVID-19 vaccination (3 studies; 9.7%). Characteristics of the included studies are summarized in Table 2 (see Appendix A for a detailed version of the table).

### 3.4. PROGRESS-Plus Health Equity Factors and Perception and/or Uptake of COVID-19 Health Information

For the 31 included studies, we mapped the PROGRESS-Plus [7] health equity factors as they relate to the uptake of COVID-19 health information and recommendations. The following factors were frequently studied and reported regarding the observational outcomes of interest: place of residence, race/ethnicity/language, occupation, gender, religion, education (health literacy/digital health literacy), socioeconomic status, social capital, and plus factors such as age, disability, and chronic illnesses.

### 3.5. Frequently Examined Health Equity Factors

The most common health equity factors examined in our selected papers on COVID-19 health information were education (E), health literacy, and digital health literacy (19/31, 61.3%). A total of 15 studies (48.4%) found that gender (G) was associated with individuals’ perceptions of and behaviors regarding COVID-19 health information. Another health equity factor influencing the perception of COVID-19 information and recommendations was age (Plus), which was suggested in 15 studies (48.4%). Socioeconomic status (S) was identified in 11 studies (35.5%) as influencing health behaviors related to COVID-19 prevention. Another important correlate reported as influencing the perception of COVID-19 information was the place of residence (P), which was examined in 10 studies (32.3%). Nine studies (29%) highlighted the significance of race, ethnicity, culture, and language (R). Additionally, one study (3.2%) reported on people with disabilities and chronic diseases (Plus).

### 3.6. Outcome Characteristics

Of the 31 included studies, 19 (61.3%) reported on perception, understanding, awareness, knowledge, adoption, and uptake of COVID-19 health information and recommendations. There were 11 studies (35.5%) that reported on intention, engagement, compliance, and adherence to COVID-19 measures and recommendations. Furthermore, 7 studies (22.6%) highlighted the role of attitudes, behaviors, and beliefs toward COVID-19 information and recommendations. There were 2 studies (6.5%) that examined information seeking and sharing health information as their outcomes of interest. Only 1 study (3%) reported on the public’s decision making about vaccination and vaccine uptake.

### 3.7. Characteristics of COVID-19 Information Content

Our findings suggested that COVID-19 messages are delivered via various pathways and in different forms or actionable statements. Four studies (13%) populated COVID-19 vaccine information and recommendations in their delivered messages. Four studies (13%) reported on COVID-19 health guidelines as an approach to inform decision making among the public. Our 31 included studies broadly examined COVID-19 information, recommendations, protective measures, and news.

## 4. Discussion

Equity factors may play an important role in the dissemination and implementation of guidelines [66,67]. Our study sheds light on the existence of evidence gaps around the relative influence of health equity factors on the understanding and uptake of COVID-19 health information and recommendations. Health literacy was a predominant term and concept in our included studies. Studies often reported on education and health literacy in relation to perceptions on COVID-19 health information. We also found that age, gender, socioeconomic status, and race/ethnicity correlated with COVID-19 attitudes and behaviors.

Our review suggests that health equity factors may be associated with susceptibility to misinformation. For example, persons with a low level of health literacy showed a preference for social media [49] and a stronger acceptance of COVID-19 misinformation circulating on social media platforms [56]. Our review also showed that COVID-19 health information sources vary across countries, ranging from local non-official information sources to official government-based organization publications. The UK DISCERN Instrument recognizes that not all health information is good-quality and that information may be inaccurate or confusing [68]. DISCERN highlights the importance of credibility of source of information to help consumers judge the quality of the information. Our review also suggests that access to credible and trusted sources of health information is essential since many platforms may disseminate misinformation, which may threaten public health [69].

Health literacy stood out as a frequently reported factor associated with the uptake and trust of information. More than half of our studies reported that perceptions of COVID-19 health information and recommendations, as well as health-related decision-making, are influenced by the level of education and health literacy. This is critical as the effectiveness of COVID-19 mitigation measures demands a comprehensive perception and support from the public [70]. This finding suggests that information sources should possibly invest in improving public understanding as well as simply communicating health information statistics. The eCOVID-19 RecMap project [26] has identified the importance of public engagement and plain language in the development of health information summaries and has developed a community process to draft and produce plain-language recommendation summaries that improve both the understanding of science and relevant scientific findings [71]. These health information “ingredients” are relevant for other health information implementation efforts.

Gender also emerged as another frequently reported health equity factor; however, there was considerable inconsistency of reported outcomes across the studies. The COVID-19 pandemic has highlighted gender disparities in health-related decision making [72]. Research suggests that the gender gaps in seeking health information and in the social and economic consequences of COVID-19 may have lessened with the passage of time in the pandemic [73]. For example, vaccination acceptance may differ by gender [51]. Efforts are needed to make COVID-19 health information understandable for the general public and people with lower health literacy and education levels. Our included studies almost exclusively focused on male and female gender, and this suggests an urgent need for research considerations for other gender types.

Race, ethnicity, or language is another health equity factor that might influence attitudes and behaviors toward COVID-19 health information and potentially result in health inequities. There are inconsistencies in outcomes across the extracted papers in relation to ethnicity. Different racial-ethnic groups of people showed different willingness to adopt the COVID-19 recommendations or get vaccinated; for example, studies showed an association between the intention to get vaccinated and different racial-ethnic groups in the population [74]. The future design of digital patient information may need to be both community participatory as well as user-centric [75]. As well, to reduce inequity related to limited English language proficiency, initiatives such as the eCOVID-19 RecMap [26] have developed multilingual plain-language recommendation summaries that could be easily accessible to both health professionals and the public [27].

COVID-19, with its higher morbidity and mortality in patients with chronic conditions, for example, disproportionately affects marginalized populations in healthcare and public health [76]. To combat virus-related pandemics with large outbreaks, equitable and fair access to the healthcare system and services is a requisite to address healthcare disparities. The Commission on Social Determinants of Health (CSDH) also accentuated the importance of addressing health inequities by recommending the development of a solid evidence base [6]. The health equity factors reported in this paper reflect an initial list of equity factors that governments could consider when formulating their policy interventions and mitigation measures.

### 4.1. Implications for Research and Knowledge Translation

Our report on 31 studies suggests a potential health equity role in the development and dissemination of digital health information and guidelines for the public. Our research suggests a role for a systematic review (including appraisal and synthesis of evidence) on the influence of health equity factors on uptake of health information. Reducing and mitigating health inequities will require community-oriented design and participation and ongoing policy and practice research.

### 4.2. Strengths and Limitations

Strengths of our approach include the use of a predefined protocol and reporting guidelines to guide this review. The reviewers independently reviewed and selected the articles based on the inclusion criteria. Nevertheless, there are some limitations to our scoping review process. We were not able to search unpublished literature, and as per the scoping review convention, reviewers did not critically appraise the quality of the evidence, limiting the depth of our interpretation. We focused on the PROGRESS-Plus framework but recognize that many other theoretical psychological and choice frameworks could be also useful in understanding the dynamic nature of how the public makes choices.

## 5. Conclusions

Our studies suggest that the public turned in to several communication channels to learn about COVID-19 prevention guidelines. The most commonly reported outcomes included perception, understanding, and uptake of information. Our results showed several gaps in the literature related to the influence of health equity factors that are worth further study. The most commonly examined health equity factor was education and health literacy, followed by gender, age, and SES. More systematic research is needed to better estimate the impact of these health equity factors on public perceptions and behaviors related to COVID-19 information and recommendations.

## Figures and Tables

**Figure 1 ijerph-19-12073-f001:**
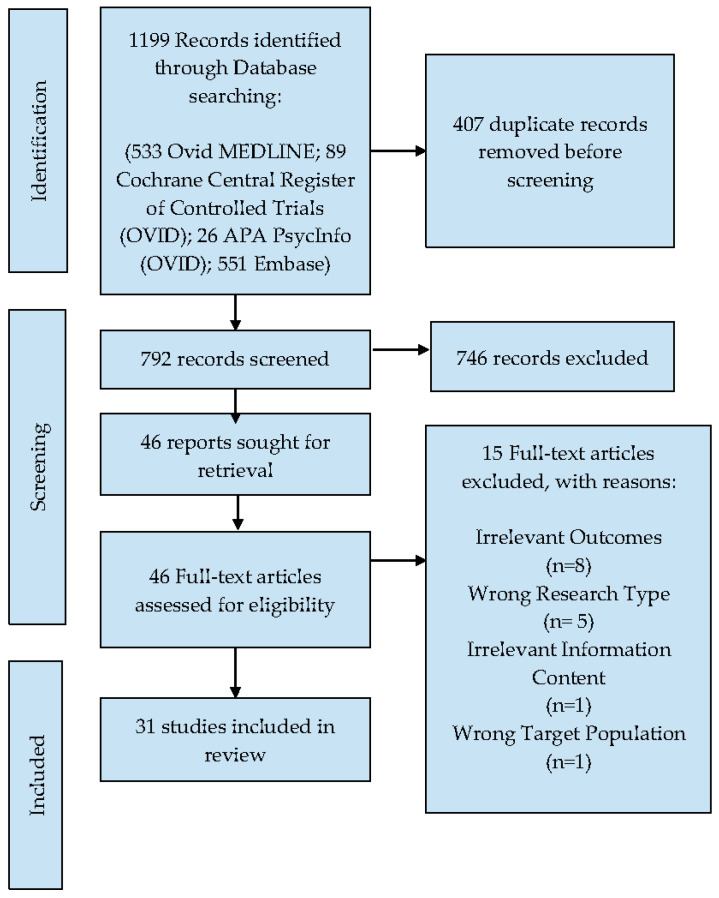
The preferred reporting items for systematic reviews and meta-analyses (PRISMA) study flow diagram.

**Figure 2 ijerph-19-12073-f002:**
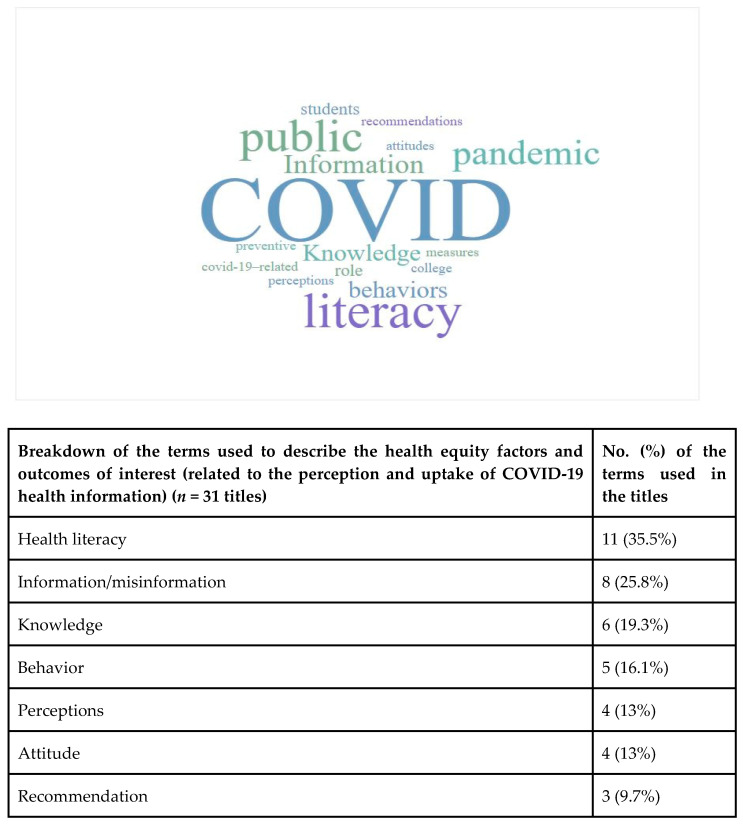
Word cloud displaying the frequency of the popular terms used in the titles of extracted papers. The font size of the word corresponds to its frequency of use in the included studies’ titles. The larger the terms, the higher the frequency. The frequency of the terms displayed on the word cloud are shown in the above table.

**Table 1 ijerph-19-12073-t001:** Selection criteria for studies included in the review.

Criteria Dimension	Inclusion Criteria	Exclusion Criteria
**Types of Participants/Population (Sample)**	General public (e.g., students, patients, caregivers, etc.)	Physicians, researchers, health professionals, academics, and other people that are not general health information users
**Exposure of Interest**	COVID-19 health information and recommendations provided by the different sources of information	Health information other than COVID-19
**Research Type**	Research publications (that have methods, data and analysis), quantitative, qualitative, or mixed-method documents published in peer-reviewed publications	Commentaries, literature reviews, gray literature
**Year of Publication**	1 January 2020–26 July 2021	Prior to 2020
**Language of Publication**	All languages	Not applicable

**Table 2 ijerph-19-12073-t002:** Characteristics of included studies, ordered alphabetically by author.

Authors/ Year	Source Country	Study Design	COVID-19 Information Content	Source of COVID-19 Health Information	Reported PROGRESS- Plus Health Equity Factors	Outcomes
Alanezi et al., 2020 [35]	Saudi Arabia	Cross-sectional survey	COVID-19-related prevention information and measures (i.e., awareness, management, myths)	(a) Ministry of Health; (b) Friends and relatives; (c) Recognized bodies such as the World Health Organization; (d) Research organizations; (e) Experts; (f) Social media; (g) Television; (h) Radio; (j) Mobiles; (k) Newspapers; (l) Community centers; (m) NGOs; (n) Local campaigns	Not reported	Uptake and awareness of COVID-19 information
Barry et al., 2020 [36]	United States of America	Cross-sectional survey	General COVID-19 related health information	(a) Social media and social circle of family and friends; (b) Internet sites and searches; (c) Third-party reports (e.g., television, radio and newspaper); (d) Scientific sources (e.g., CDC and professional journals)	Education/health literacy, socioeconomic status	Knowledge and uptake of COVID-19 health recommendations
Basch et al., 2021 [37]	United States of America	Successive sampling (longitudinal study)	COVID-19 vaccination recommendations	(a) Social media (e.g., YouTube)	Not reported	Decision making about COVID-19 vaccination and vaccine uptake
Bazaid et al., 2020 [38]	Saudi Arabia	Cross-sectional survey	COVID-19 preventive health recommendations	Not reported	Place of residence, gender, education, socioeconomic status, age	Knowledge and adherence to COVID-19 preventive behaviors
Block et al., 2020 [39]	United States of America	Cross-sectional survey	COVID-19 public health prevention recommendations	Not reported	Race	Adherence to COVID-19 public health recommendations
Chen et al., 2020 [40]	China	Cross-sectional survey	General COVID-19 health information and preventive recommendations	(a) Newspapers/magazines; (b) TV; (c) Radio; (d) Cellphone text messages; (e) Web portals; (f) Social media; (g) News websites; (h) Video-sharing social networking services; (j) Online question and answer platforms; (k) Search engines; (l) Online learning platforms	Place of residence, education/health literacy, socioeconomic status, age	Engagement in preventive behaviors, behavioral intention, attitude, subjective norms, knowledge, interpersonal sources of information, media sources of information, information appraisal
Czeisler et al., 2020 [41]	United States of America	Cohort survey	COVID-19 prevention guidelines (stay-at-home orders, masks, physical distancing, group gathering, inside dining, self-isolation).	Not reported	Place of residence, occupation, age	Attitudes, behaviors, and beliefs related to COVID-19 preventive health guidelines
Desalegn et al., 2021 [42]	Addis Ababa, Ethiopia	Cross-sectional survey	COVID-19 prevention guideline	(a) Government-owned television; (b) Government-owned radio; (c) Social media; (d) Privately-owned television	Occupation	Knowledge, attitude, practice, and engagement in recommended prevention behaviors
Enria et al., 2021 [43]	United Kingdom	Cross-sectional survey	COVID-19 prevention health information	(a) United Kingdom government reports	Place of residence, race, gender, education, socioeconomic status, age	Uptake and acceptance of COVID-19 preventive measures by the government
Hermans et al., 2021 [44]	Belgium	Cross-sectional survey	COVID-19 preventive recommendations	Not reported	Education/health literacy	Compliance with COVID-19 preventive measures
Kerr et al., 2021 [45]	United Kingdom	Experimental surveys (Trials)	COVID-19 vaccine information	(a) US Food and Drug Administration; (b) European Medicines Agency; (c) Centres for Disease Control and Protection; (d) British Society for Immunology; (e) Pfizer	Not reported	Uptake of COVID-19 vaccine and information
Kor et al., 2021 [46]	China Hong Kong Macau	Cross-sectional survey	General COVID-19-related prevention information	(a) Search engines; (b) Websites of public bodies; (c) Wikipedia and other online encyclopedias; (d) Social media; (e) YouTube; (f) Blogs on health topics; (g) Online communities; (h) Health portals	Gender, education/health literacy/digital health literacy, socioeconomic status, age, disability	Perception of information and satisfaction
Lennon et al., 2020 [47]	United States of America	Cross-sectional survey	COVID-19 public health recommendations	Not reported	Place of residence, race	Knowledge, perceptions, preferred health information sources, and understanding of and intent to comply with public health recommendations
Li, Shaojie et al., 2021 [48]	China	Cross-sectional survey	COVID-19 prevention recommendations	(a) Guidelines for Public Protection Against Pneumonia Caused by the Novel Coronavirus Infection, which were compiled by the Chinese Center for Disease Control and Prevention (China CDC); (b) Internet; (c) Media; (d) Social media	Place of residence, gender, education/health literacy/eHealth literacy (digital health literacy), socioeconomic status	Adoption of COVID-19-related preventive behaviors
Li, Yingkai et al., 2020 [49]	United States of America	Prospective observational cohort study	COVID-19 prevention information and recommendations	(a) Federal government, state governments; (b) local healthcare providers; (c) television news, and presidential news; (d) conferences and addresses; (e)websites, social media; (f) religious organizations	Religion, education, socioeconomic status, age	Knowledge and perceptions of COVID-19 information and recommendations
McCaffery et al., 2020 [50]	Australia	Cross-sectional survey	COVID-19 prevention and vaccine information	(a) Public television; (b) Social media; (c) Government websites	Race, gender, education/health literacy, age	Knowledge, attitudes, beliefs, behaviors, and uptake of COVID-19 information
Montagni et al., 2021 [51]	France	Prospective cohort survey	COVID-19 vaccine information and recommendations	(a) Media; (b) Social media	Gender, education/health literacy	Uptake of information or misinformation (i.e., detection of fake news)
Motta Zanin et al., 2020 [52]	Italy and abroad	Case study	General COVID-19-related prevention information and COVID-19 recommendations	(a) Television; (b) Social networks; (c) Newspapers; (d) Internet; (e) Scientific journals; (f) Radio; (g) Relatives and friends; (h) General practitioners; (j) Other	Not reported	Perception of mitigation measures
Ng et al., 2021 [53]	United States of America	Cross-sectional survey	COVID-19 prevention information and recommendations	(a) Traditional news sources, including television, radio, websites, and newspapers; (b) Social media; (c) Comments or guidance from government officials; (d) Other webpages/Internet; (e) Friends or family members; (f) Health care providers	Place of residence, race, gender, socioeconomic status, social capital, age	Engagement in recommended prevention behaviors
Okan et al., 2020 [54]	Germany	Cross-sectional survey	General COVID-19 prevention information	(a) Internet; (b) Newspapers; (c) Magazines; (d) Television	Place of residence, gender, education/health literacy, socioeconomic status, age	COVID-19 health recommendations access, understanding, appraisal, and applying
Patil et al., 2021 [55]	United States of America	Cross-sectional survey	General COVID-19-related information	(a) Internet; (b) Social media	Race, gender, education/health literacy/digital health literacy, disability	COVID-19-related information access, attitudes, and behaviors
Pickles et al., 2021 [56]	Australia	Prospective longitudinal cohort study	COVID-19 prevention information and recommendations	(a) Family and friends; (b) Television; (c) Radio; (d) Print media; (e) Health (and allied) care providers	Race, gender, education/digital health literacy, age	Perception and uptake of information or misinformation
Riiser et al., 2020 [57]	Norway	Cross-sectional survey	General COVID-19 prevention information and protective measures	(a) TV; (b) Radio; (c) Newspapers; (d) Podcasts; (e) YouTube, Snapchat, TikTok, Instagram, Facebook, other media; (f) Family and friends; (g) School	Place of residence, gender, education/health literacy	Knowledge, behavior, and uptake of COVID-19 health information
Rose et al., 2021 [58]	United States of America	Cross-sectional survey	COVID-19 health recommendations	Not reported	Race, gender, education, socioeconomic status, age	Compliance with COVID-19 preventive measures
Schafer et al., 2021 [59]	Germany	Cross-sectional survey	General COVID-19-related health information	(a) Medical online consultation (e.g., online consultation of doctors or hospitals); (b) Online radio, audio streaming and/or podcast; (c) Online television and/or video streaming (e.g., Netflix, Amazon Prime); (d) Online pharmacies, comparison portals for searching doctors, hospitals, and nursing homes (e) Websites of (non-profit) health organizations, independent patient or self-help organizations; (f) Service communities; (g) Websites of health insurance companies; (h) Heath forums and Communities specialized in health and disease issues; (j) Social media; (k) Blogs on health and disease; (l) Websites of doctors, hospitals, rehabilitation or care institutions; (m) Video platforms; (n) Online news sites; (o) Health portals; (p) Wikipedia or other online encyclopedias; (q) Search engines	Gender, health literacy, age	Seek COVID-19 health recommendations, perception, and behavior
Schultz et al., 2021 [60]	France	Cross-sectional survey (Short Communication)	General scientific advice and COVID-19 public health policies	Not reported	Not reported	Perception of COVID-19 public health policies
Syropoulos et al., 2021 [61]	United States of America	Cross-sectional Study	General COVID-19 prevention information and guidelines	Not reported	Not reported	Adhering to health guidelines
Tang et al., 2021 [62]	China	Semistructured interviews (case study)	COVID-19 prevention information, personal opinions, and outbreak information	(a) Governmental organizations and state media; (b) Social media, personal accounts and group chats, media, TV; (c) Online news subscription services, news websites, and online newspapers and search engines	Education, age	Information seeking, scanning, and sharing (health information consumption)
VanScoy et al., 2021 [63]	United States of America	Cross-sectional survey	General COVID-19-related prevention information and behavioral recommendations	(a) Government websites (e.g., Centres for Disease Control and Prevention, National Institutes of Health [NIH], and the World Health Organization [WHO]); (b) Television news	Race, gender, education, age	Adherence, knowledge, understanding of public health recommendations, perceptions, and trust in information sources related to COVID-19
Vardavas et al., 2021 [64]	G7 Countries (Canada, France, Great Britain, Germany, Italy, Japan, and the United States)	Cross-sectional survey	Governmental communication regarding prevention guidelines and the COVID-19 outbreak	(a) Doctors/health care providers; (b) Friends/family; (c) Government/politicians; (d) Mass media (e.g., newspapers/news websites/television); (e) Social media (e.g., Facebook, Twitter)	Place of residence	Engagement in COVID-19 preventive behaviors
Wong et al., 2020 [65]	Hong Kong	Cross-sectional survey	General COVID-19 health information	(a) Family members; (b) Social media	Gender, education/health literacy, socioeconomic status, age	Engagement in preventive behaviors and COVID-19 information sharing

**Table 3 ijerph-19-12073-t003:** Mapping PROGRESS-Plus health equity factors on perception and uptake of health information.

Health Equity PROGRESS-Plus Factors	Mapping Health Equity Factors Related to the Perception and Uptake of COVID-19 Health Information and Behaviors
Place of Residence (P)	(Bazaid et al., 2020, Saudi Arabia [38]): Youth residents of the northern and western regions of Saudi Arabia reported lower preventive behavior practice scores than youth residents in other regions. (Chen et al., 2020, China [40]): Rural residents had more negative attitudes toward regulations, were less likely to appraise health information, and were less likely to take COVID-19 preventive measures. There was no difference between the knowledge of preventive behaviors, subjective norms, and behavioral intentions. There was no difference between rural and urban residents in interpersonal/media source use. (Czeisler et al., 2020, USA [41]): Nationwide, respondents from urban areas reported using cloth face coverings at a higher percentage than respondents from rural regions. (Enria et al., 2021, United Kingdom [43]): Participants from the East of England, the Southeast, and the West Midlands all had higher odds than Londoners of thinking that the government was making good decisions. (Lennon et al., 2020, United States [47]): Respondents from various cities of the US showed differing levels of intent to comply with the health recommendations. (Li, Shaojie et al., 2021, China [48]): There were different prevention behavior scores among different residents. (Ng et al., 2021, USA [53]): Residents of metro areas indicated a greater probability of engaging in all three preventive behaviors. (Okan et al., 2020, Germany [54]): There was no significant difference between people from different regions in confusion about COVID-19. (Riiser et al., 2020, Norway [57]): Compared to rural residents, urban people had a greater engagement in social distancing. (Vardavas et al., 2021, G7 countries [64]): There were different levels of approval and trust in government responses to the pandemic based on the place of residence.
Race, Ethnicity, Culture, and Language (R)	(Block et al., 2020, USA [39]): African Americans were less likely (other populations were not assessed) to follow fundamental public health guidelines (e.g., handwashing). (Enria et al., 2021, United Kingdom [43]): Ethnic minorities were unequally influenced by COVID-19 regulations. (Lennon et al., 2020, United States [47]): White women showed better health outcomes. (McCafferey et al., 2020, Australia [50]): When compared to people who predominantly spoke English at home, those who reported speaking a language other than English (LOTE) at home rated the threat of COVID-19 as lower, with a larger proportion reporting that they were not likely to get sick. Compared to those who spoke English at home, people who spoke an LOTE at home had a harder time recognizing COVID-19 symptoms and infection-prevention measures. (Ng et al., 2021, USA [53]): Hispanic individuals and those speaking languages other than English at home engaged in more preventive behaviors compared to non-Hispanic white individuals. (Patil et al., 2021, USA [55]): No difference was detected for health literacy across ethnic or racial groups. (Pickles et al., 2021, Australia [56]): People speaking languages other than English at home tended to agree more with misinformation statements. (Rose et al., 2021, USA [58]): Being in a racial/ethnic minority group was related to greater comparative compliance, higher general intentions, and lower risk perceptions. (VanScoy et al., 2021, USA [63]): Non-minority race patients reported significantly higher knowledge.
Occupation (O)	(Czeisler et al., 2020, USA [41]): Essential workers reported lower adherence to recommendations for self-isolation, physical distance, and restricting gatherings. (Desalegn et al., 2021, Ethiopia [42]): Occupational status had a more positive attitude toward preventive measures.
Gender/Sex (G)	(Bazaid et al., 2020, Saudi Arabia [38]): Females were shown to have higher adherence scores. (Enria et al., 2021, United Kingdom [43]): Males showed less trust in government decisions. (Kor et al., 2021, Hong Kong and China and Macau [46]): Males reported lower health information seeking. (Li, Shaojie et al., 2021, China [48]): Females showed more vigilance in COVID-19 precautionary behaviors. (McCafferey et al., 2020, Australia [50]): Females showed more difficulty in perceiving government messages. (Montagni et al., 2021, France [51]): The study observed a significant association between sex and the intention to receive the COVID-19 vaccine. Data (numbers and percentages) are provided for males and females for the categories of anti-vaccination, hesitancy, and pro-vaccination. (Ng et al., 2021, USA [53]): Compared to males, females showed more engagement in preventive behaviors. (Okan et al., 2020, Germany [54]): Females reported more confusion about COVID-19. (Patil et al., 2021, USA [55]): Students identified as female or gender-variant reported more adequate health literacy than males. (Pickles et al., 2021, Australia [56]): Males showed a higher tendency to follow misinformation. (Riiser et al., 2020, Norway [57]): The gender differences were significant in terms of engagement in preventive behaviors. Females were more likely to indicate compliance to protective measures than males. (Rose et al., 2021, USA [58]): Females reported greater comparative and absolute compliance, higher general intentions, higher worry and risk perceptions, and greater severity perceptions. (Schafer et al., 2021, Germany [59]): No significant differences were evident between female students, male students, and students who identified themselves as diverse. (VanScoy et al., 2021, USA [63]): There was no difference between males and females in COVID-19 knowledge level. (Wong et al., 2020, Hong Kong [65]): Females were associated with personal preventive behaviors while living in public housing.
Religion (R)	(Li, Yingkai et al., 2020, USA [49]): Although religious organizations were considered as one of the sources people use to gather information regarding COVID-19, there is no evidence to support that differences in religious beliefs lead to differences in perception, uptake, and attitudes toward COVID-19 health information.
Education (Health Literacy/Digital Health Literacy) (E)	(Barry et al., 2020, USA [36]): Even after controlling for patients’ characteristics, patients with better general health literacy and education had better knowledge of basic epidemiology, prevention, diagnosis, treatment, and prognosis of COVID-19. (Bazaid et al., 2020, Saudi Arabia [38]): People with lower levels of education had lower adherence scores in protective measures. (Chen et al., 2020, China [40]): Urban people show higher levels of education, which is directly related to higher acceptance of preventive behaviors and information appraisal. (Enria et al., 2021, United Kingdom [43]): Higher education leads to less trust in government responses and decisions. (Hermans et al., 2021, Belgium [44]): Participants with adequate health literacy have a lower risk of not adhering than those with low health literacy. (Kor et al., 2021, Hong Kong and China and Macau [46]): Less educated people reported less frequent health information seeking. (Li, Shaojie et al., 2021, China [48]): Higher levels of eHealth literacy were associated with greater conventional health behaviors, suggesting that college students in China with greater eHealth literacy could maintain healthy lives throughout the COVID-19 pandemic. (Li, Yingkai et al., 2020, USA [49]): People with higher levels of education showed lower trust in social media information. (McCafferey et al., 2020, Australia [50]): Individuals with low health literacy were less likely to have changed their plans and less likely to perceive social distancing as necessary. Still, compared to people with adequate health literacy, they were far more likely to feel personally unprepared for a large outbreak. Since the lockdown, people with low health literacy have had more trouble remembering and getting medications. In general, people with lower health literacy were more likely than those with adequate health literacy to endorse inaccurate beliefs regarding COVID-19 and vaccinations. (Montagni et al., 2021, France [51]): When compared to those with a high health literacy level, those with a low health literacy score were more likely to be “hesitant” rather than “pro-vaccination”. (Okan et al., 2020, Germany [54]): Confusion was not associated with educational level. (Patil et al., 2021, USA [55]): Higher digital health literacy was significantly associated with a higher willingness to vaccinate against COVID-19 and a belief that contracting the disease would negatively impact their lives. (Pickles et al., 2021, Australia [56]): People with a lower level of education represent more agreement with the misinformation. (Riiser et al., 2020, Norway [57]): Health literacy levels were associated with different preventive measures. Literate people showed a higher tendency to follow the health authorities’ guidelines. (Rose et al., 2021, USA [58]): Higher education level was associated with lower behavior-specific intentions. (Schafer et al., 2021, Germany [59]): There was a direct relation (positive association) between health literacy and intensity of seeking health information. (Tang et al., 2021, China [62]): Similar statements (trust in information sources) came from all educational levels. (VanScoy et al., 2021, USA [63]): Knowledge of COVID-19 public health recommendations was significantly higher in patients with higher education. (Wong et al., 2020, Hong Kong [65]): Health literacy and COVID-19 information sharing with family members were associated.
Socioeconomic Status (S)	(Barry et al., 2020, USA [36]): After controlling for confounding variables, this study found that emergency department patients with a low income had a lower level of COVID-19 knowledge. (Bazaid et al., 2020, Saudi Arabia [38]): People from low socioeconomic origins had less knowledge about COVID-19 transmission and lower adherence scores. (Chen et al., 2020, China [40]): Different behavioral intentions and knowledge between rural and urban people come from the difference in this SES determinant. (Enria et al., 2021, United Kingdom [43]): Participants with lower income levels reported less positive opinions of the government’s decisions related to COVID-19 and a lower likelihood of following COVID-19 health recommendations. (Kor et al., 2021, Hong Kong and China and Macau [46]): People with lower socioeconomic status reported less frequent health information seeking. (Li, Shaojie et al., 2021, China [48]): There were different prevention behavior scores among different economic levels (higher COVID-19-specific precautionary behavior scores were positively associated with middle economic level versus low economic level). (Li, Yingkai et al., 2020, USA [49]): People from different socioeconomic backgrounds did not show diverse levels of trust in various information sources. Furthermore, there is no difference between these individuals’ concerns, caution, or anxiety levels. (Ng et al., 2021, USA [53]): People with lower income reported lower odds of engaging in all three preventive behaviors. (Okan et al., 2020, Germany [54]): Participants with higher income felt more informed about COVID-19. (Rose et al., 2021, USA [58]): Higher income was associated with greater comparative compliance. (Wong et al., 2020, Hong Kong [65]): Higher personal income was associated with personal preventive behaviors while living in public housing.
Social Capital (SS)	(Ng et al., 2021, USA [53]): People living alone have less tendency to engage in preventive behaviors compared to people not living alone.
Age (Plus)	(Bazaid et al., 2020, Saudi Arabia [38]): Four different age groups (18–27; 28–37; 38–47; above 47 years) participated in this study; those under the age of 37 had lower adherence scores in protective measures. (Chen et al., 2020, China [40]): Younger people represented fewer preventive behaviors and lower intention to engage in these behaviors, and were more likely to have a negative attitude toward behavioral change. (Czeisler et al., 2020, USA [41]): Younger people might feel safer without community mitigation strategies (might relate to perceived risk for infection and severe disease). (Enria et al., 2021, United Kingdom [43]): Younger participants in the survey showed less trust in the government’s actions and guidelines (COVID-19 responses). (Kor et al., 2021, Hong Kong and China and Macau [46]): Older-aged respondents reported less frequent health information seeking. (Li, Yingkai et al., 2020, USA [49]): Older people showed lower trust in the uptake of health information from social media. (McCafferey et al., 2020, Australia [50]): Younger people showed more difficulty in perceiving government messages. (Ng et al., 2021, USA [53]): Younger beneficiaries reported higher odds of engaging in all three preventive behaviors. (Okan et al., 2020, Germany [54]): Younger people showed more confusion about COVID-19. (Pickles et al., 2021, Australia [56]): Younger people were more vulnerable to misinformation. (Rose et al., 2021, USA [58]): Older participants indicated greater comparative compliance, higher general intentions, and greater severity perceptions. (Schafer et al., 2021, Germany [59]): Younger people were more keen to use social media platforms to find health information. (Tang et al., 2021, China [62]): Younger people showed more reliance on social media and the internet, while older people showed a higher propensity to using TV as their primary source for information. (VanScoy et al., 2021, USA [63]): There was no difference between ages in COVID-19 knowledge level. (Wong et al., 2020, Hong Kong [65]): Older age was associated with higher health literacy.
Disability and chronic illnesses (Plus)	(Kor et al., 2021, Hong Kong and China and Macau [46]): The majority of people with chronic diseases used social media to find online information about COVID-19, and their level of satisfaction with that information was significantly lower than that of people without chronic diseases. (Patil et al., 2021, USA [55]): People with disability showed lower digital health literacy. (Schafer et al., 2021, Germany [59]): Students with chronic diseases sought health information significantly more often than students without such diseases.

## Data Availability

The Appendix A holds information extracted from studies used in this review.

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
