# Peer review of "Identifying Health Equity Factors That Influence the Public’s Perception of COVID-19 Health Information and Recommendations: A Scoping Review"

_ijerph, 2022, doi:10.3390/ijerph191912073_

Round 1

Reviewer 1 Report

This manuscript addresses a relevant topic in public health: the role of health inequalities in the public acceptance of control measures during the covid-19 pandemic. A scoping review based on the Progress-Plus Health inequity guide is performed to identify the most relevant social determinants of health in the population's perception of the public health recommendations. 

The manuscript is clear and well written. However, there are several limitations to be addressed:

First, the authors point out that they focus on perceptions, but the results also include outcomes as engagement, adherence, and compliance with the control measures (i. e, lines 210-218, line 232, Table 3). This point should be clarified along the manuscript, from the objective, the search strategy and the results, to the conclusions.

Second, a possible bias is perceived towards overestimating health literacy as a determinant of health. In table 2, health literacy is reported in 9 studies (although educational factor summarises 18 studies), whereas gender was identified in 15 studies, the same as age. However, gender and age do not stand out in the results as decisive factors in the perception and adherence to protective recommendations. The authors should clarify the differences between the definition of education and health literacy, especially in those primary studies which include both variables.

Methodology

-The authors may justify the reason of the end period of publications searched: Why until July 26, 2021?

-Line 132: Although the authors point out that in the scoping review it is not necessary to assess the quality of the studies, the PRISMA-ScR guide does recommend that a critical appraisal of and within individual sources of evidence should be carried out. A quality assessment could help to interpret the findings. 

-Line 134: The methodology to map the word cloud, specifically the related terms chosen, should be detailed to assess possible bias towards an overestimation of the effect of health literacy over the rest of health determinants. 

-As well, the name of the software and its version used should be included in the text. 

 Results:

-Figure 1: In the flow chart, reasons to exclude 746 articles should be detailed inside the box.

-Unlike the data shown in the abstract, the proportion of studies according to the methodology (cohort studies, experimental studies, case studies, qualitative studies using interviews, etc.) is lacking in the results section.

-Table 2: ‘Digital health literacy’ may be included as an inequity factor, as the rol of this factor is highlighted by the authors in future research and knowledge translation.

Discussion

-A comment about differences in the findings in each type of methodological design (i.e., cross sectional studies compared to cohort or experimental studies) would be appreciated to discuss the scope and possible bias to produce evidence on this topic.

Conclusion

-Line: 319-321: The first two sentences are not in correspondence to the main research question. 

-Line 323: Education and health literacy are highlighted as the main health equity factor. This must be in accordance with the results.

Reviewer 2 Report

This paper provides a useful summary and meta analysis of studies that focus on how trust in public-health provided COVID-19 prevention information changed from Jan 2020 through Jul 2020. The authors find evidence that equity factors were a strong contributor to public trust of COVID-19 prevention intervention. The paper is well written and does a good job of documenting the various attributes of the literature they include in their analysis. I recommend publication.  

Reviewer 3 Report

Well done on your paper! 

Overall, an important paper highlighting the significance of equity in health, and the importance of addressing the local population needs.

Below are some suggestions on how the paper could be strengthened. I hope you find this useful. 

Introduction

Line 50-52: There is a currently established link between poor health literacy and poor outcomes. Those with poor health literacy have lower engagement with health services and lower ability to self-manage care.

Line 62- 63: Is the direct quotation used appropriately referenced? Consider checking.

Methods

Line 99-100: It is mentioned that searches were conducted in databases without language restrictions. Please provide more details such as was translation of papers required?  What language papers were translated from and who conducted this?

Line 108: For clarity, if possible, please state age bracket of the ‘youth’ and ‘seniors’ population included in the study.

Line 114: In the ‘Criteria Dimension’ column, consider removing the word “Intervention” and leave criteria as “Exposure of Interest” as the wording may not be appropriate to all study designs considered in this paper.

Results

Line 150: Replace the number ‘31’ with the word ‘thirty-one’ for consistency.

Line 154: Check Figure 1 – some words are not displayed.  Consider re-submitting the image.

Line 154: Please clarify what is meant by ‘wrong research type’ (n=5)? Do you mean these papers were not peer reviewed?

Lines 180-185: Important to state number of papers and percentages of papers that did not report on the source of COVID-19 Health Related Information.

Lines 180-185: Are the percentages based on papers that only reported the source of COVID-19 Health Related Information or the total number of papers? Please clarify. This comment also applies to papers that reported PROGRESS-Plus Health Equity Factors.

Line 226: Table 3:

§  Third row “Occupation” for the Desalegn et al 2021 Ethiopia study, please explicitly state occupational status had a more positive attitude toward preventive measures if possible.

§  Fourth row “Gender/sex” column, for the Montagni et al 2021 France study, please explicitly state the observed association between sex and intention to receive the vaccine if possible. 

§  Sixth row “Education (Health Literacy/Digital Health Literacy)” for the Riiser et al 2020 Norway study, please give an example of health literacy levels that were associated with different preventive measures if possible. 

Implications for Research and Knowledge Translation

§  It would be helpful if the authors could offer recommendations on how these findings could be utilised by individuals, community groups, clinicians, and policy makers.

All the best. 
